**Data Availability Statement:** Data cannot be shared publicly because of local ethics committee decision. Data are available from the Ege University

# Contribution of "complete response to treatment" to survival in patients with unresectable metastatic colorectal cancer: A retrospective analysis

Gulcan Bulut[1]*, Merve Guner Oytun[2], Elvina Almuradova[3], Mustafa Harman[4], Ruchan Uslu[5], Bulent Karabulut[3]

1 Division of Medical Oncology, Defne Hospital, Antakya, Hatay, Turkey, 2 Division of Geriatric Medicine, Department of Internal Medicine, Hacettepe University Medical School, Ankara, Turkey, 3 Division of Medical Oncology, Department of Internal Medicine, Ege University Medical School, Izmir, Turkey, 4 Department of Radiology, Ege University Medical School, Izmir, Turkey, 5 Division of Medical Oncology, Department of Internal Medicine, Celal Bayar University Medical School, Manisa, Turkey

* gulcanbulut07@gmail.com

## Abstract

### Background

The aim of the study is to reveal the contribution of complete response (CR) to treatment to overall survival (OS) in patients with unresectable metastatic colorectal cancer. In addition, to evaluate progression-free survival (PFS) in patients who attained CR to treatment and to examine the clinicopathologic features of the patient group with CR.

### Methods

This article is a retrospective chart review. Patients diagnosed with metastatic colorectal cancer were divided into two groups. The systemic treatment was compared with the patients who received a full response according to the Response Evaluation Criteria in Solid Tumors (RECIST1.1) and those who did not attain CR (progression partial response and stable response) in terms of both PFS and OS data, and the effect of attaining CR to treatment on prognosis was evaluated.

### Results

A total of 222 patients were included in the study. 202 of 222 patients could be evaluated in terms of complete response. All data from their files were tabulated and analyzed retrospectively. The mean age of diagnosis of the study group was 60.13 ± 12.52 years. The total number of patients who attained CR to treatment was 31 (15.3%); 171 (84.6%) patients did not attain CR. Patients who had a CR had longer median PFS times than patients who did not have a CR (15.2 vs. 7.4 months, **P<0.001**). Patients who had CR had longer median survival times than patients who did not have a CR (39.2 vs. 16.9 months, **P<0.001**). In subgroup patients who underwent primary surgery, the number of patients who attained CR was statistically higher compared with the number of patients who did not attain CR

Medical School,Izmir-Turkey Data Access / Ethics Committee (contact via:egetaek@gmail.com) for researchers who meet the criteria for access to confidential data.

**Funding:** Unfunded studies: The author(s) received no specific funding for this work.

**Competing interests:** The authors have declared that no competing interests exist.

(**p<0.001**). Complete response was less common in the presence of liver metastasis and bone metastasis (**p = 0.041 and p = 0.046**, respectively), had a negative prognostic effect. In other words, 89.1% of patients with liver metastasis, 100.0% of patients with bone metastasis, and 88.7% of those who died did not have a CR to the treatment. According to multivariate analysis, CR to treatment, primary surgery, first-line chemotherapy (combination compared with fluoropyrimidine), and no bone metastasis were found to be predictors for OS.

## Conclusion

Providing CR with systemic treatment in patients with unresectable metastatic colorectal cancer (mCRC) contributes to prognosis. The primary resection in our secondary acquisitions from the study, the number of metastatic regions and the combination therapy regimens also contributed to the prognosis.

## Introduction

Colorectal cancer (CRC) is the third most diagnosed malignancy in Europe, with an estimated 447,000 new cases diagnosed annually in Europe [1]. CRC is the second most common cause of cancer-related deaths in the Western world; 20% of patients have been found to have metastatic colorectal cancer (mCRC) at diagnosis [1]. Overall survival (OS) for patients with unresectable metastatic colorectal cancer with "best supportive care" (BSC) is 6 months [2]. In studies in the literature, the OS achieved with systemic therapy in patients with mCRC is approximately 2 years [3, 4]. Accordingly, the fact that the comparison parameter is BSC in new randomized studies is unethical and patients with metastatic colorectal cancer are directed to systemic therapy [5, 6]. Increasing OS with the development of biologic agents arouses excitement in clinical practice. Terminology of maximum tumor response revealed by the "First-line therapy for patients with metastatic colorectal cancer" (FIRE-3) study is used in the clinical evaluation of patients in terms of predicting survival. In studies, the depth of response was proven to contribute to OS with the FIRE-3 study [7].

In the literature, it is known that systemic treatment results in patients with mCRC, complete response (CR) is obtained in 10–15% of patients according to RECIST1.1 criteria as known as "Response Evaluation Criteria in Solid Tumors" [8]. RECIST 1.1 is a radiologic common evaluation language developed for clinical studies with a primary endpoint objective response. Although the importance of response depth was demonstrated in the FIRE-3 study, information about patients with CRs is limited [9, 10]. In Turkey, there are no similar studies in the literature.

In our study, it was aimed to compare patients who attained a CR according to RECIST1.1 in the report of radiology and nuclear medicine as a result of systemic treatment with a diagnosis of unresectable mCRC compared with patients who did not attain a CR in terms of clinicopathologic features, treatment regimen, progression-free survival (PFS) and OS data.

## Materials and methods

All procedures performed in studies involving human participants were conducted in accordance with the ethical standards of the institutional research committee and with the 1964

Helsinki Declaration and ethical standards. (Ege University Clinical Research Ethics Committee approval number: 16–4.4/4).

The files of patients diagnosed with metastatic colorectal cancer who applied to the outpatient clinics of Ege University Medical Oncology Department between January 1st, 2007, and December 31st, 2015, were analyzed retrospectively. And the date of termination of follow up January 1st, 2020. Patients aged over 18 years and diagnosed with metastatic colorectal cancer, regardless of whether they received adjuvant chemotherapy (CT), were included in the study. In other words, both patients with de-novo metastasis and those who developed metastasis after primary tumor resection and adjuvant therapy were included in the study. Patients who underwent metastasectomy at any stage of the treatment, except for biopsies taken for pathologic sampling, patients with another cancer and patients with life-threatening comorbidity were excluded from the study. Age, primary disease region, primary diagnosis date, primary surgery date if the primary tumor is operated, pathologic features of the primary tumor (degree, location of the tumor in the wall of the intestine wall, regional lymph node status = TNM staging), KRAS status, presence of adjuvant treatment, previous primary disease adjuvant treatments (such as primary tumor surgery, adjuvant CT, neo-adjuvant radiotherapy), metastasis history, metastasis regions, first-line and subsequent treatments, start and end dates of treatments, best radiologic response with the treatment applied, last visit date, and date of death if the patient was deceased were saved to the SPSS program.

In addition, the imaging of patients whose radiology evaluations were given in full was reevaluated by the radiology department according to the 'Evaluation Criteria in Solid Tumors' (RECIST1.1) and CR was confirmed in a single center. Thus, the difference between centers and CR was standardized.

## Statistical analysis

Statistical analyses were performed using the IBM SPSS Statistics Version 22 package program. Apart from applying descriptive statistics to the data, the Chi-square test, Student's t-test, and the Mann-Whitney U test were used in subgroup comparisons, and Kaplan-Meier and Cox regression analyses were used in survival analyses. The statistical limit of significance was accepted as $p<0.05$. The whole study was a retrospective file data evaluation and no test, examination and/or intervention was performed.

## Results

### Patient characteristics

A total of 222 patients were included in the study. The mean age of diagnosis of the study group was 60.13 ± 12.52 years. The study group consisted of 134 (60.4%) men and 88 (39.6%) women. The median patient follow-up time was 19.07 (min: 0.3 months-max: 131.6) months. The mean metastasis interval was 5.93 ± 12.81 months. The mean number of metastatic regions was 1.69 ± 0.1. The primary involvement site of the tumor was rectal 82 (37.6%), distal colon 81 (37.2%), proximal colon 44 (20.2%) and transverse colon in 11 (5.0%) patients. Primary surgery was not performed in 80 (40.6%) patients, whereas it was performed in 132 (59.4%) patients. Thus, patients with de novo metastasis comprised 40.6% of the study group. Forty-eight (36.3%) patients received adjuvant therapy with patients with primary surgery. This group might be associated with chemoresistance. Histopathologic diagnosis of the tumor was adeno cancer in 184 (83.6%) patients. In tumor staging, 69 (47.3%) patients were in T3, 48 (32.9%) patients were in T4. Fifty-one (71.8%) patients had lymphovascular invasion (LVI), and 39 (63.2%) patients had perineural invasion (PVI). In terms of KRAS status, 65 (29.5%) patients had mutant-type and 76 (34.5%) patients had wild-type. One hundred forty-two

(64.5%) patients had liver metastasis, and 22 (9.9%) patients had bone metastasis. The total number of patients who attained a CR to treatment was 31 (14.6%); 181 (85.4%) patients did not attain a CR. The number of patients with progression was 172 (97.7%). Twenty (9.0%) patients were still alive, 202 (91.0%) patients had died.

## Factors associated with attaining complete response

The comparison of clinical and histopathologic findings according to the response to treatment is shown in Table 1. 202 of 222 patients could be evaluated in terms of complete response. The mean age of patients who attained CR was 57.3 ± 12.6 years, and the mean age of patients who did not attain CR was 60.5 ± 12.2 years. This data was not statistically significant (**p = 0.192**). In patients who underwent primary surgery, the number of patients who attained CR was 28 (21.9%); the number of patients who did not attain CR was 100 (78.1%). In patients who did not undergo primary surgery, the number of patients who attained CR was 3 (3.8%), and the number of patients who did not attain CR was 71 (96.3%). These data were statistically significant (**p<0.001**). There was a statistically significant difference between patients who attained CR in terms of no liver metastasis, no bone metastasis, and survivor status compared with those who did not attain CR (**p = 0.041, p = 0.046, and p<0.001, respectively**). In other words, 89.1% of patients with liver metastasis, 100.0% of patients with bone metastasis and 88.7% of those who died did not have a CR to treatment (**Table 1**).

When the tumor location subgroup analysis was performed, there was no statistically significant difference between the tumor locations (**p = 0.072**). The rates of patients with rectum cancer and colon cancer who attained CR was 8.9% and 18.3%, whereas the rates of the same cancers in who did not attain a CR were 91.1% and 81.7%.

**Survival and predictors of survival and complete response (CR).** According to the multivariate analysis (logistic regression), the effects of variables on the "CR to treatment" are shown in Table 2. According to the logistic regression analysis, performing primary surgery significantly affects the CR to treatment (**p = 0.006**, Hazard Ratio HR: 0.168, 95% CI: 0.047–0.594). The effects of the age of diagnosis, liver metastasis, bone metastasis and first-line CT (fluoropyrimidine, irinotecan, and oxaliplatin) parameters on the CR to treatment were not statistically significant.

OS analysis based on whether CR was attained is shown in Table 2. Patients who had a CR had longer median survival times than patients who did not have a CR (39.2 vs. 16.9 months, **P≤0.001**) (**Fig 1, Table 2**).

PFS analysis, based on whether a CR was attained is shown in Table 3. Patients who had a CR had longer median PFS than patients who did not have a CR (15.2 vs. 7.4 months, **P<0.001**) (**Fig 2, Table 3**).

According to the univariate analysis (Cox regression analysis), the effect of variables on OS is shown in Table 4. According to the Cox regression analysis, it was found that performing primary surgery contributed to a statistically significant increase in OS (**p<0.001**, HR: 1.719, 95% CI: 1.282–2.305). It is understood that performing primary surgery increases the OS by 1.719 times. Patients who undergo primary surgery have longer OS than patients with de novo metastasis. The decreases in the number of metastatic regions statistically significantly increased the OS time (**p<0.001**, HR: 1.375, 95% CI: 1.178–1.604). It is understood that a one-unit increase in the number of metastatic regions reduces OS by 1.375 times. When examining the metastasis sites, two groups were noteworthy. One of these was liver, the most common site of metastasis, and bone metastasis, one of the rare metastatic sites. Having liver metastases significantly reduced OS (**p = 0.031**, HR: 0.721, 95% CI: 0.535–0.971). Having bone metastases significantly reduced OS (**p<0.001**, HR: 0.396, 95% CI: 0.250–0.627). It was found that first-

**Table 1. The comparison of clinical and histopathologic findings according to the complete response to treatment.**

| | Complete Response + | | Complete Response - | | P |
|---|---|---|---|---|---|
| | n = 31 Mean±SD | % Min-max | n = 171 Mean±SD | % Min-max | |
| Age at diagnosis | 57.3±12.6 | | 60.5±12.2 | | 0.192 |
| Sex | | | | | 0.731 |
| Male | 18 | 14.0 | 111 | 86.0 | |
| Female | 13 | 15.7 | 70 | 84.3 | |
| Metastasis interval (month) | 6.5±11.7 | | 5.8±13.2 | | 0.767 |
| Number of metastatic regions | 1 (1.09–1.62) | | 1 (1.60–1.87) | | - |
| Primary region | | | | | 0.160 |
| Rectum | 7 | 8.9 | 72 | 91.1 | |
| Distal colon | 15 | 19.0 | 64 | 81.0 | |
| Proximal colon | 6 | 14.3 | 36 | 85.7 | |
| Transverse colon | 3 | 30.0 | 7 | 70.0 | |
| Primary surgery | | | | | **<0.001** |
| No | 3 | 3.8 | 71 | 96.3 | |
| Yes | 28 | 21.9 | 100 | 78.1 | |
| Histology | | | | | 0.364 |
| Unknown | 1 | 9.1 | 10 | 90.9 | |
| Adenocarcinoma | 25 | 14.0 | 153 | 86.0 | |
| Mucinous adenocarcinoma | 5 | 29.4 | 12 | 70.6 | |
| Signet ring cell adenocarcinoma | 0 | 0.0 | 3 | 1.0 | |
| Neuroendocrine differentiated adenocarcinoma | 0 | 0.0 | 3 | 1.0 | |
| Differentiation | | | | | 0.460 |
| Good | 2 | 18.2 | 9 | 81.8 | |
| Middle | 21 | 21.6 | 76 | 78.4 | |
| Poor | 1 | 6.3 | 15 | 93.7 | |
| Unknown | 7 | 8.0 | 81 | 92.0 | |
| T stage | | | | | - |
| T1 | 1 | 100.0 | 0 | 0.0 | |
| T2 | 1 | 100.0 | 0 | 0.0 | |
| T3 | 16 | 25.0 | 48 | 75.0 | |
| T4 | 9 | 19.1 | 38 | 80.9 | |
| Tx | 0 | 0.0 | 26 | 100.0 | |
| LVI | | | | | 0.303 |
| Yes | 13 | 26.0 | 37 | 74.0 | |
| No | 7 | 38.9 | 11 | 61.1 | |
| PNI | | | | | 0.461 |
| Yes | 9 | 24.3 | 28 | 75.7 | |
| No | 7 | 33.3 | 14 | 66.7 | |
| KRAS status | | | | | 0.759 |
| Unknown | | | | | |
| Mutant | 10 | 15.9 | 53 | 84.1 | |
| Wild type | 12 | 16.0 | 63 | 84.0 | |
| Metastasis time | | | | | 0.786 |
| At the time of diagnosis | 24 | 15.3 | 133 | 84.7 | |
| After adjuvant treatment | 7 | 13.7 | 44 | 86.3 | |

(*Continued*)

**Table 1.** (Continued)

| | Complete Response + | | Complete Response - | | P |
|---|---|---|---|---|---|
| | n = 31 Mean±SD | % Min-max | n = 171 Mean±SD | % Min-max | |
| Liver metastasis | | | | | **0.041** |
| No | 16 | 21.3 | 59 | 78.7 | |
| Yes | 15 | 10.9 | 122 | 89.1 | |
| Bone metastasis | | | | | **0.046** |
| No | 31 | 16.2 | 149 | 83.8 | |
| Yes | 0 | 0.0 | 22 | 100.0 | |
| Adjuvant therapy | | | | | 0.609 |
| No | 23 | 14.0 | 141 | 86.0 | |
| Yes | 8 | 17.0 | 39 | 83.0 | |
| First-line treatment | | | | | 0.276 |
| Fluoropyrimidine | 1 | 5.6 | 17 | 94.4 | |
| Irinotecan | 12 | 19.4 | 50 | 80.6 | |
| Oxaliplatin | 17 | 13.1 | 113 | 86.9 | |
| Survival status | | | | | **<0.001** |
| Survivor | 9 | 52.9 | 8 | 47.1 | |
| Died | 22 | 11.3 | 173 | 88.7 | |

LVI: Lymphovascular invasion, PNI: Perineural invasion.

line treatment with fluoropyrimidine and a first-line treatment combination with irinotecan significantly increased OS (**p = 0.007**, **p = 0.003**, respectively). First-line treatment with combination treatment compared with single-agent fluoropyrimidine was found to increase OS statistically significantly (**p = 0.002**, HR: 2.196, 95% CI: 1.339–3.600). It was found that attaining a CR to treatment contributed to the most positively statistically significant increase in OS (**p<0.001**, HR: 2.648, 95% CI: 1.691–4.147) (**Table 4**).

According to the multivariate analysis (logistic regression), the effects of variables on OS are shown in Table 5. In the multivariate analysis, CR to treatment (**p = 0.002**, HR: 2.107, 95% CI: 1.324–3.353); first-line CT (combination compared with single-agent fluoropyrimidine) (**p = 0.014**, HR: 1.909, 95% CI: 1.140–3.199); performing primary surgery (**p = 0.033**, HR: 1.406, 95% CI: 1.028–1.924) were found to be positively significantly associated with OS. These parameters prolonged OS and did not cross the line in the HR chart. It is understood that the most positively effective factor on OS is the "CR to treatment" status. Having bone metastases was found to be negatively associated with OS (p = 0.01 HR:0.488 95% CI: 1.140–3.199). Unlike the univariate analysis, decreases in the number of metastatic regions were not found to be significantly associated with OS (**p = 0.108**, HR: 1.17, 95% CI: 0.996–1.416). And unlike the univariate analysis, having liver metastases was not found to be negatively significantly

**Table 2.** Overall survival analysis based on whether a complete response was attained.

| OS | Median (month) | 95% C.I. | | P |
|---|---|---|---|---|
| | | Lower | Upper | |
| **CR -** | 16.990 | 14.702 | 19.278 | |
| **CR +** | 39.230 | 22.141 | 56.319 | **<0.001** |
| Total | 19.580 | 16.646 | 22.514 | |

CR: Complete response to treatment, OS: Overall Survival.

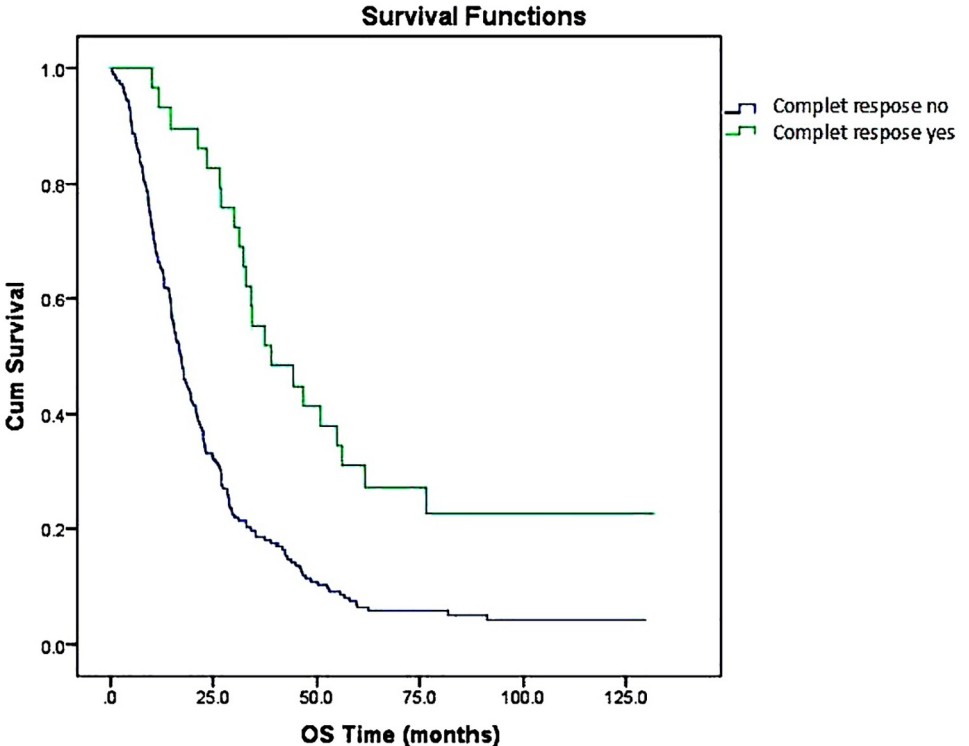

**Fig 1. Overall survival analysis based on whether a complete response was attained.**

associated with OS (**p = 0.265**, HR: 0.830, 95% CI: 0.283–0.840). Only bone metastases short-ened OS but did not cross the line on the HR chart (**Fig 3**, **Table 5**).

## Discussion

In this retrospective study, we investigated the clinicopathologic features, treatment regimen, OS, and PFS of patients with unresectable mCRC who attained a CR with systemic treatment compared with patients without a CR. It is known that a CR is obtained in 10–15% of patients with metastatic colorectal cancer who receive systemic treatment according to the RECIST1.1 criteria [8]. In our study, similar to the literature, 15.3% of our group of 202 patients attained a CR. Having CR to treatment was found to be associated with OS and PFS. According to the univariate analysis, performing primary surgery, decreased numbers of metastatic regions, liver metastasis, no bone metastasis, first-line CT (fluoropyrimidine), first-line CT (irinote-can), and CR to treatment were found to be associated with OS. However, in the multivariate analysis, primary surgery, first-line CT (combination compare with fluoropyrimidine alone)

**Table 3. Progression-free survival analysis based on whether a complete response was attained.**

| PFS | Median (month) | 95% CI | | P |
|---|---|---|---|---|
| | | Lower | Upper | |
| **CR -** | 7.490 | 6.088 | 8.892 | |
| **CR +** | 15.240 | 13.126 | 17.354 | **<0.001** |
| Total | 8.800 | 7.422 | 10.178 | |

CR: Complete response to treatment, PFS: Progression-free Survival.

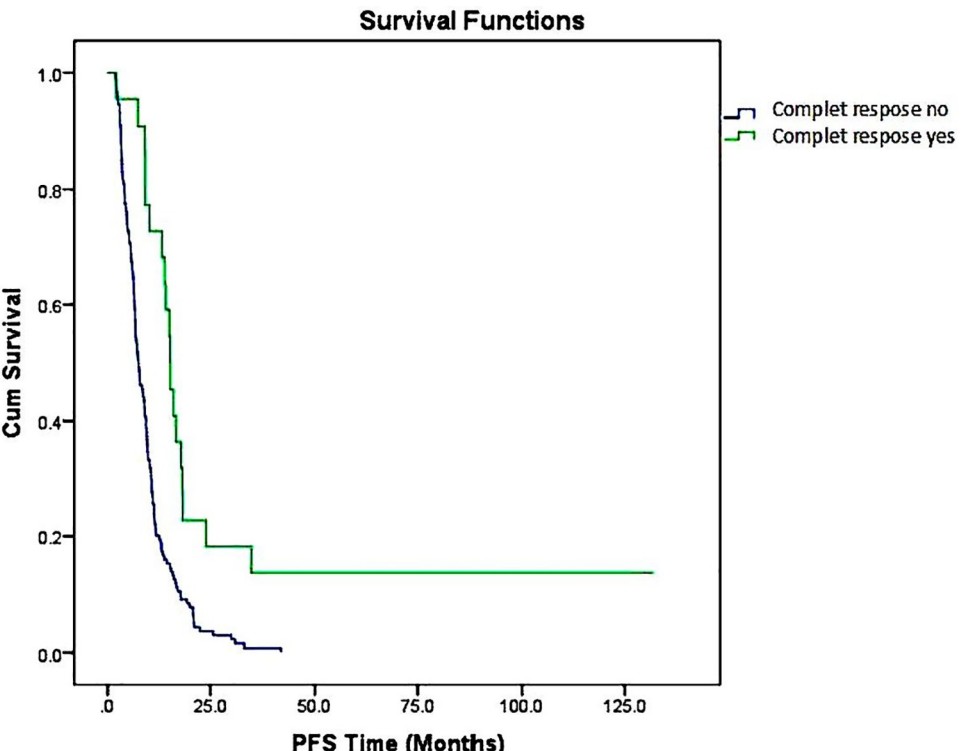

**Fig 2. Progression-free survival analysis based on whether a complete response was attained.**

and CR to treatment were found to be positively associated with OS, and bone metastasis was negatively associated with OS.

CRC is the third most common cancer diagnosed between both sexes with an estimated death rate of 51,020 and an estimated 145,600 new cases for 2019 [11]. Metastatic CRC (mCRC) is an advanced age malignancy that presents at a median age of 67 years [12]. In our study, the mean age of diagnosis of mCRC was 60.13 ± 12.52 years, younger than in the literature. Significantly, OS and PFS number of outliers was significantly lower with increasing age [13]. Contrary to the literature, age was not found to be prognostic for OS and PFS in our study.

mCRC occurs in >50% of cases, the majority of which are inoperable at presentation [14, 15]. In our study, patients with de novo metastasis comprised 40.6% of the study group, slightly less than in the literature. Both patients with de novo metastasis and patients who had primary tumor surgery and received neoadjuvant/adjuvant therapy were included in the study. It was found that performing primary surgery contributed to a statistically significant increase in OS. De novo metastatic colorectal cancer was associated with poor prognosis.

Approximately 60% of patients with CRC will develop liver metastases [16]. Most patients have a recurrence after partial hepatectomy, but approximately 20% are cured [17, 18]. However, the vast majority (80–90%) presents unresectable diseases. In the study of D´Angelica et al. on 49 colorectal cancer patients with unresectable hepatic metastasis was reported that only four patients achieved a CR after CT (oxaliplatin, irinotecan, fluoropyrimidine, and bevacizumab). The median OS was 38 months. The median OS and PFS for all patients were 38 and 13 months. CR was obtained in 10 of 49 patients after primary surgery. The median OS of these patients was 39 months. They reported that primary surgery and treatment with modern systemic CT of colorectal-liver metastasis (CRLM) were associated with long-term survival

**Table 4. According to the univariate analysis (Cox regression analysis), the effect of variables on overall survival.**

| | Exp(B) | 95.0% CI for Exp(B) | | P |
|---|---|---|---|---|
| | | Lower | Upper | |
| Age of diagnosis | 1.007 | 0.995 | 1.019 | 0.265 |
| Sex (Male to female) | 0.824 | 0.618 | 1.100 | 0.190 |
| Primary region (Rectum) | | | | 0.726 |
| Primary region (Distal colon) | 0.690 | 0.367 | 1.300 | 0.252 |
| Primary region (Proximal colon) | 0.722 | 0.382 | 1.365 | 0.316 |
| Primary region (Transverse colon) | 0.724 | 0.370 | 1.416 | 0.345 |
| **Primary surgery (performed according to not performed)** | **1.719** | **1.282** | **2.305** | **<0.001** |
| Histology (Adeno) | | | | 0.178 |
| Histology (Mucinous) | 0.470 | 0.149 | 1.488 | 0.199 |
| Histology (Signet ring) | 0.367 | 0.106 | 1.274 | 0.114 |
| Histology (Neuroendocrine) | 1.147 | 0.231 | 5.694 | 0.866 |
| Differentiation (Good) | | | | 0.316 |
| Differentiation (Medium) | 0.585 | 0.259 | 1.323 | 0.198 |
| Differentiation (Poor) | 0.662 | 0.375 | 1.171 | 0.156 |
| T (T1) | 0.717 | 0.097 | 5.301 | 0.744 |
| T (T2) | 2.483 | 0.332 | 18.590 | 0.376 |
| T (T3) | 0.615 | 0.385 | 0.981 | 0.042 |
| T (T4a - T4b) | 0.790 | 0.479 | 1.302 | 0.356 |
| The number of LAP excised | 1.000 | 0.988 | 1.012 | 0.992 |
| Metastatic LAP number | 1.025 | 0.997 | 1.054 | 0.082 |
| LVI (According to the non-existent) | 0.729 | 0.415 | 1.283 | 0.273 |
| PNI (According to the non-existent) | 0.648 | 0.358 | 1.174 | 0.152 |
| KRAS status (mutant according to Wild) | 0.851 | 0.603 | 1.202 | 0.361 |
| Metastasis time (after adjuvant treatment according to the time of diagnosis) | 0.990 | 0.715 | 1.371 | 0.952 |
| Disease-free interval | 0.996 | 0.986 | 1.007 | 0.470 |
| **Decreases in the number metastatic regions** | **1.375** | **1.178** | **1.604** | **<0.001** |
| **Liver metastasis** | **0.721** | **0.535** | **0.971** | **0.031** |
| **Bone metastasis** | **0.396** | **0.250** | **0.627** | **<0.001** |
| Neoadjuvant therapy (according to the non-existent) | 0.898 | 0.269 | 2.995 | 0.861 |
| Adjuvant therapy (according to the non-existent) | 1.062 | 0.757 | 1.490 | 0.729 |
| First-line CT (Fluoropyrimidine)* | | | | **0.007** |
| First-line CT (İrinotecan) | 2.146 | 1.296 | 3.554 | **0.003** |
| First-line CT (Oxaliplatin) | 0.931 | 0.678 | 1.279 | 0.660 |
| **First-line CT (Combination with Fluoropyrimidine)** | **2.196** | **1.339** | **3.600** | **0.002** |
| **CR to treatment** | **2.648** | **1.691** | **4.147** | **<0.001** |

CT: Chemotherapy, LVI: Lymphovascular invasion, PNI: Perineural invasion, LAP: Lymphadenopathy, CR: Complete Response to treatment.

* Complete response in the first-line CT (Fluoropyrimidine) group is only one patient. Therefore, other values in the row could not be presented.

[19]. In our study, 64.5% of patients with mCRC had liver metastasis. The number of patients with liver metastases and CRs was 15. Similar to the literature, in our study, patients who had a CR had longer median survival times than patients who did not have a CR (39.2 vs. 16.9 months). Patients who had a CR had longer median PFS than patients who did not have a CR (15.2 vs. 7.4 months).

Bone metastasis is a rare consequence of colorectal cancer and is a sign of poor prognosis. Several reports in the literature describe a positive response to double CT, with targeted therapy currently being the standard treatment [20]. In a study by Arslan et al., the authors

**Table 5. According to the multivariate analysis (logistic regression), the effects of variables on overall survival.**

|  | Exp(B) | 95.0% CI for Exp(B) | | P |
|---|---|---|---|---|
|  |  | Lower | Upper |  |
| **Primary surgery (performed according to not performed)** | **1.406** | **1.028** | **1.924** | **0.033** |
| Liver metastasis | 0.830 | 0.598 | 1.152 | 0.265 |
| **Bone metastasis** | **0.488** | **0.283** | **0.840** | **0.010** |
| Decreases in the number metastatic regions | 1.170 | 0.966 | 1.416 | 0.108 |
| **First-line CT (Combination with Fluoropyrimidine)** | **1.909** | **1.140** | **3.199** | **0.014** |
| **CR to treatment** | **2.107** | **1.324** | **3.353** | **0.002** |

CT: Chemotherapy, CR: Complete Response to treatment.

detected bone metastasis at the 4th month during the treatment of FOLFOX plus bevacizumab for a patient with unresectable mCRC. The patient died 4 months after developing bone metastasis. They stated that bone metastases could be the precursor of short survival [21]. In a study by Nakamura et al., a 51-year-old patient with mCRC with bone metastasis died 16.6 months after the first-line CT treatment [20]. In our study, 22 (9.9%) patients had bone marrow metastases. In terms of bone marrow metastases, there is a statistical significance in patients who attained a CR to treatment compared with those who did not attain a CR. Patients with bone metastasis were not able to attain a CR to treatment. According to the univariate analysis (Cox regression analysis), it was found that the bone metastases significantly reduced the OS (p<0.001, HR: 0.396, 95% CI: 0.250–0.627). In the patient group with bone metastasis, the median OS was 16.990 months, which was quite low. In the multivariate analysis, bone metastasis was found to be significantly associated with OS (p = 0.010, HR: 0.488, 95% CI: 0.283–0.840). None of the patients with bone metastases had a CR to treatment. Bone metastases in metastatic colorectal cancer was also associated with poor prognosis. With bone targeted

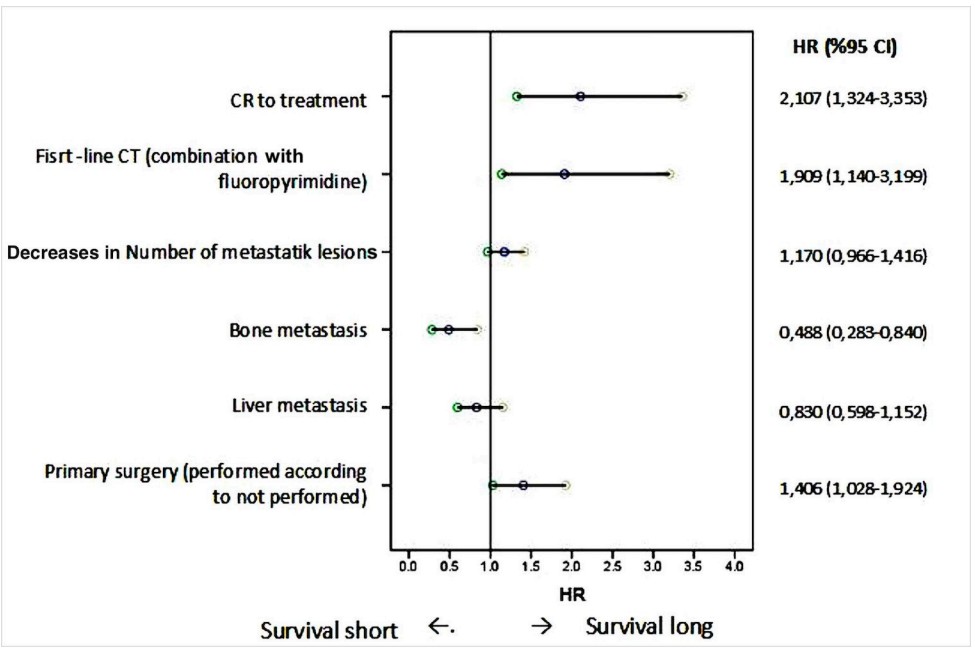

**Fig 3. According to the multivariate analysis (logistic regression), the effects of variables on overall survival.**

drugs and radiotherapy methods that can be applied locally to the bone, a complete response may be possible in bone metastasis. Setting new targets could be useful for drug development studies in such patients with poor prognostic subgroups of mCRC.

Treatment of unresectable mCRC is mitigated with systemic antineoplastic therapy aimed at maintaining OS, symptom management, and quality of life [14]. The median survival of patients with advanced colorectal cancer treated with single fluoropyrimidine is approximately 1 year, whereas it is approximately 2 years in those treated with a combination of irinotecan or oxaliplatin with fluoropyrimidine-based CT. As stated in the literature, our patients were receiving fluoropyrimidine, irinotecan, and oxaliplatin treatment as first-line CT. However, it was also found that first-line CT treatment with fluoropyrimidine and irinotecan significantly increased OS, but not first-line CT with oxaliplatin. The combination of new targeted agents (bevacizumab, cetuximab or panitumumab) and CT have extended life expectancy of patients with metastatic colorectal cancer over 30 months and 5-year survival in certain mCRC subsets was reported as 14.2% [14, 15]. In our study, first-line treatment was evaluated regardless of the use of biological agents. Even so, this is what makes our article original; patients who had a CR to treatment had longer median survival times than patients who did not have a CR to treatment (39.2 vs. 16.9 months). Patients who had a CR to treatment had longer median PFS times than patients who did not have a CR (15.2 vs. 7.4 months). CR significantly improved the prognosis. Nevertheless after the long follow-up time, our study reported only 8.2% of patients were alive.

Despite the strengths of the current study, some limitations should be taken into account when considering the results. First, as with any retrospective study, there was a bias inherent in its nature. Second, some studies have identified performance condition as an important prognostic factor in mCRC. However, we did not have access to these data for inclusion in our study. Third, we also did not have information about what treatments the patient received after the first-line treatment in patients who could not achieve CR, maybe sequential systemic treatments could be examined and the incidence of CR could be increased. Fourth, in the sub-group analysis of patients with rectum cancer, the number of patients who received neoadjuvant radio chemotherapy could not be analyzed because there were only six patients. The last limitation, perhaps the most important limitation, is that the study was conducted with rather small number of patients. Future prospective and large database studies should be used to further validate and investigate our results. Interestingly, the CR was an independent predictor of survival for about 40months in OS and for about 15 months in PFS. This series also advises that there could be a higher rate of patients with mCRC attaining CR with raised targets and targeted therapy.

## Conclusion

In conclusion, the primary results of our study showed that patients with unresectable mCRC who attained CR had significantly prolonged survival with a permanent response to therapy. The secondary result of our study showed that de novo metastasis or liver metastasis or bone metastasis in patients who did not attain CR was associated with poor prognosis. Finally, the study provides some important results that physicians can use in terms of expected life expectancy when counseling patients who respond completely.

## Supporting information

**S1 File.**
(DOCX)

## Author Contributions

**Data curation:** Merve Guner Oytun.

**Formal analysis:** Mustafa Harman.

**Investigation:** Merve Guner Oytun, Elvina Almuradova, Mustafa Harman.

**Methodology:** Gulcan Bulut, Elvina Almuradova.

**Project administration:** Ruchan Uslu.

**Supervision:** Ruchan Uslu, Bulent Karabulut.

**Validation:** Bulent Karabulut.

**Visualization:** Gulcan Bulut, Bulent Karabulut.

**Writing – original draft:** Gulcan Bulut.

**Writing – review & editing:** Ruchan Uslu, Bulent Karabulut.

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
