## [Decision Letter · Decision Letter 0]

10 Dec 2020

PONE-D-20-28477

Contribution of “Complete Response to Treatment” to Survival in Patients with Unresectable Metastatic Colorectal Cancer; Retrospective Analysis

PLOS ONE

Dear Dr. Bulut,

Thank you for submitting your manuscript to PLOS ONE. After careful consideration, we feel that it has merit but does not fully meet PLOS ONE’s publication criteria as it currently stands. Therefore, we invite you to submit a revised version of the manuscript that addresses the points raised during the review process.

We look forward to receiving your revised manuscript.

Kind regards,

Nader Hanna

Academic Editor

PLOS ONE

Additional Editor Comments:

The manuscript needs through review/edits for the English language and grammar.

Journal Requirements:

2. In the ethics statement in the manuscript and in the online submission form, please provide additional information about the patient records used in your retrospective study, including: a) whether all data were fully anonymized before you accessed them and/or whether the IRB or ethics committee waived the requirement for informed consent; b) the date range (month and year) during which patients' medical records were accessed. If patients provided informed written consent to have data from their medical records used in research, please include this information.

3. We noted at least one instance of p=0.000 in your manuscript. Please report  exact p-values for all values greater than or equal to 0.001. P-values less than 0.001 may be expressed as p < 0.001. For more information on PLOS ONE's expectations for statistical reporting, please see https://journals.plos.org/plosone/s/submission-guidelines.#loc-statistical-reporting.

4. We suggest you thoroughly copyedit your manuscript for language usage, spelling, and grammar. If you do not know anyone who can help you do this, you may wish to consider employing a professional scientific editing service.  

5. During your revisions, please confirm whether the wording in the short title is correct and update it in the manuscript file and online submission information if needed. Specifically, the current short title on the submission form includes "Running title:", which is redundant. Additionally, the short title does not provide much information about the content of the manuscript.

Reviewers' comments:

Reviewer's Responses to Questions

**Comments to the Author**

1. Is the manuscript technically sound, and do the data support the conclusions?

Reviewer #1: Partly

2. Has the statistical analysis been performed appropriately and rigorously? 

Reviewer #1: No

3. Have the authors made all data underlying the findings in their manuscript fully available?

Reviewer #1: No

4. Is the manuscript presented in an intelligible fashion and written in standard English?

Reviewer #1: No

5. Review Comments to the Author

Reviewer #1: There are multiple grammatical and typographical errors throughout the manuscript that make it difficult to easily read. These should be reviewed prior to resubmission to improve readability of manuscript.

The authors performed a retrospective review of patients diagnosed with unresectable Stage IV colorectal cancer at a single institution to determine how different treatment strategies affected patient outcomes (overall survival, progression free survival), based on whether patients did or did not attain complete response (CR) to treatment. The authors report that patients who underwent primary surgery had a higher rate of obtaining CR. Patients with liver metastasis and bone metastasis were unlikely to obtain CR. Patients with CR had longer median survival times and progression free survival. Additionally, patients who underwent first line treatment with fluoropyrimidine or irinotecan also increased overall survival, while number of metastatic regions reduced overall survival. While I believe this study and results are interesting and important, the data is presented in somewhat confusing fashion and the conclusions that are drawn are not always clearly presented.

Authors should include in the abstract the type of study this was (a retrospective chart review).

There are multiple places in the manuscript were it is stated p=0.0000. While this may be correct from a rounding standpoint, from a statistical standpoint, I believe it would be more correct to state p<0.001.

What exactly do the authors mean by "primary surgery"? In the Methods section, they state that any patients who underwent metastectomy during treatment were excluded. Does "primary surgery" indicate surgery as the FIRST step in their treatment? Or rather that the patient underwent surgery of their primary tumor? What surgeries did these patients undergo (resection, debulking, diversion) etc, and what were the indications for these operations?

Colon and rectal cancers are grouped together in this analysis, but I do not see any mention of radiation treatment for patients with rectal cancer. Was this not included in their neoadjuvant treatment? It be interesting to see a subgroup analysis of colon vs rectal cancer to ID any difference between the two groups.

In discussing the significant results from Table 2 - the way in which the data are presented is very confusing and the authors extrapolate new percentages within the text (albiet correct percentages based on number of patients with liver/bones mets or those who did not survive) to discuss the significant findings, but these are different than the percentages in Table 2. Would reword this to make the understanding more clear.

In discussing results from Table 6 - the authors interpretation of hazard ratios inconsistently increase or decrease OS (i.e. HR 1.7 for primary surgery increases OS, but HR of 1.375 for met regions DECREASES OS, same for liver vs bone mets). If this is based on how the data was analyzed, it should be described in such a fashion that this is understandable, otherwise, it appears that the results have been interpreted incorrectly.

In discussing results from Table 7 - the authors should discuss the different hazard ratios and how they each affect overall survival. The way that it is written groups all the variables together without describing their affect (i.e. surgery and chemo increase OS, while bone mets decreases overall survival).

In the discussion section, these results are again grouped together without delineating positive or negative association with the multiple variable discussed.

The discussion session includes discussion of mCRC in elderly patients, but it is unclear to me exactly how this relates to the data presented in this study?

What additional therapeutic approaches to metastatic disease are the authors referencing in the last paragraph of the Discussion section?

6. PLOS authors have the option to publish the peer review history of their article (what does this mean?). If published, this will include your full peer review and any attached files.

Reviewer #1: No

---

## [Author Response · Author response to Decision Letter 0]

13 Jan 2021

Dear Dr. Editor and Reviewers;

  

1. The article's titles and figure suggestions were arranged in accordance with PLOS ONE style. 

2.The requested correction for ethical approval was rewritten in the first paragraph of the material method section.

3. P-values less than 0.001 was expressed as p < 0.001. 

4. The article was edited with a professional scientific editing service.  

 5. the desired correction has been made

   Response to the Reviewers  1. Is the manuscript technically sound, and do the data support the conclusions?  The manuscript must describe a technically sound piece of scientific research with data that supports the conclusions. Experiments must have been conducted rigorously, with appropriate controls, replication, and sample sizes. The conclusions must be drawn appropriately based on the data presented.

Reviewer #1: Partly

The statistical mistakes , table and explanation part of the study were corrected and correct interpretation was made. 

2. Has the statistical analysis been performed appropriately and rigorously?

Reviewer #1: No

This article was received professional support for statistics.The statistics mistake and misinterpretations were corrected and presented.

3. Have the authors made all data underlying the findings in their manuscript fully available? 

Reviewer #1: No

 We've added the maniscpripts statistic data as part of the manuscript or its supporting information, or we've allowed to deposited to a public repository.

4. Is the manuscript presented in an intelligible fashion and written in standard English? 

Reviewer #1: No

The article was edited with a professional scientific editing service.

5. Review Comments to the Author  Please use the space provided to explain your answers to the questions above. You may also include additional comments for the author, including concerns about dual publication, research ethics, or publication ethics. (Please upload your review as an attachment if it exceeds 20,000 characters)

Reviewer # 1: There are multiple grammatical and typographical errors throughout the manuscript that make it difficult to easily read. These should be reviewed prior to resubmission to improve readability of manuscript. The authors performed a retrospective review of patients diagnosed with unresectable Stage IV colorectal cancer at a single institution to determine how different treatment strategies affected patient outcomes (overall survival, progression free survival), based on whether patients did or did not attain complete response (CR ) to treatment. The authors report that patients who underwent primary surgery had a higher rate of obtaining CR. Patients with liver metastasis and bone metastasis were unlikely to obtain CR. Patients with CR had longer median survival times and progression free survival. Additionally, patients who underwent first line treatment with fluoropyrimidine or irinotecan also increased overall survival, while number of metastatic regions reduced overall survival. While I believe this study and results are interesting and important, the data is presented in somewhat confusing fashion and the conclusions that are drawn are not always clearly presented. Authors should include in the abstract the type of study this was (a retrospective chart review). There are multiple places in the manuscript were it is stated p = 0.0000. While this may be correct from a rounding standpoint, from a statistical standpoint, I believe it would be more correct to state p <0.001. What exactly do the authors mean by "primary surgery"? In the Methods section, they state that any patients who underwent metastectomy during treatment were excluded. Does "primary surgery" indicate surgery as the FIRST step in their treatment? Or rather that the patient underwent surgery of their primary tumor? What surgeries did these patients undergo (resection, debulking, diversion) etc, and what were the indications for these operations? Colon and rectal cancers are grouped together in this analysis, but I do not see any mention of radiation treatment for patients with rectal cancer. Was this not included in their neoadjuvant treatment? It be interesting to see a subgroup analysis of colon vs rectal cancer to ID any difference between the two groups.

Answers;

The article was editing by David Frances Chapman

The findings and discussion section presented as confusing was rewritten.

‘a retrospective chart review ‘ was added in abstract section 

P-values less than 0.001 was expressed as p < 0.001.

Patients who received adjuvant therapy were not excluded in the study, therefore patients who underwent primary tumor resection were included in our study. This was explained in the article more clearly, necessary additions were made. Colon and rectum sub group analyzes were performed in patients and this subgroup analysis was added as a paragraph. But neoadjuvant treatment was not evaluated because only 6 of the rectal cancer patients received neoadjuvant rt. This is our limitation and was added last paragragh.

In discussing the significant results from Table 2 - the way in which the data are presented is very confusing and the authors extrapolate new percentages within the text (albiet correct percentages based on number of patients with liver / bones mets or those who did not survive) to discuss the significant findings, but these are different than the percentages in Table 2. Would reword this to make the understanding more clear

The percentages were rewritten correctly in the article. Mistakes ; It was presented in its corrected form.

 In discussing results from Table 6 - Since the p value is accepted as 0.05. in univariate analyzes the number of metastatic regions also both bone met and kc metastasis shortens OS time. But multivariete analises, only bone metastases were found to be statistically significant. While bone negative os contribution was significant in fig 3 according to HR, kc and numbers of metastasis region cut the line, so it was not significant. 

In univariate analysis, we realized and corrected our error presented by shortening it with the one without bone metastases and presented it correctly.

……….

How all the data in table 7 affects the OS was added to the article. the explanation was expressed more clearly.

………

Much information given with CRC with elderly patients was removed and restricted to the age paragraph only.

In the discussion section, these results are again grouped together without delineating positive or negative association with the multiple variable discussed. The discussion session includes discussion of mCRC in elderly patients, but it is unclear to me exactly how this relates to the data presented in this study? What additional therapeutic approaches to metastatic disease are the authors referencing in the last paragraph of the Discussion section?

The statistical mistakes , table and explanation part of the study were corrected and correct interpretation was made.Additional therapeutic approaches were actually describing sequential CT. In other words, secondary or third line treatments were not evaluated in this study. It was written into the article in a more descriptive way.

---

## [Decision Letter · Decision Letter 1]

14 Apr 2021

PONE-D-20-28477R1

Contribution of “Complete Response to Treatment” to Survival in Patients with Unresectable Metastatic Colorectal Cancer; A Retrospective Analysis

PLOS ONE

Dear Dr. Bulut,

Thank you for submitting your manuscript to PLOS ONE. After careful consideration, we feel that it has merit but does not fully meet PLOS ONE’s publication criteria as it currently stands. Therefore, we invite you to submit a revised version of the manuscript that addresses the points raised during the review process.

In agreement with the reviewer the authors did improve the manuscript, however, its presentation and readability remain weak points. Unless the presentation is improved, the manuscript may not be accepted in the present form. I would like to encourage authors to answer all comments and put an effort in improving the language. 

We look forward to receiving your revised manuscript.

Kind regards,

Ludmila Vodickova, M.D., PhD

Academic Editor

PLOS ONE

Journal Requirements:

Reviewers' comments:

Reviewer's Responses to Questions

**Comments to the Author**

1. If the authors have adequately addressed your comments raised in a previous round of review and you feel that this manuscript is now acceptable for publication, you may indicate that here to bypass the “Comments to the Author” section, enter your conflict of interest statement in the “Confidential to Editor” section, and submit your "Accept" recommendation.

Reviewer #1: (No Response)

2. Is the manuscript technically sound, and do the data support the conclusions?

Reviewer #1: Yes

3. Has the statistical analysis been performed appropriately and rigorously? 

Reviewer #1: Yes

4. Have the authors made all data underlying the findings in their manuscript fully available?

Reviewer #1: Yes

5. Is the manuscript presented in an intelligible fashion and written in standard English?

Reviewer #1: No

6. Review Comments to the Author

Reviewer #1: Informed consent was obtained from all 222 patients in the study? This is not typical of a retrospective chart review.

There are still multiple grammatical and typographical errors throughout the paper that make it somewhat difficult to read, for example (this does not include all the errors in the article, just a sampling):

Abstract

Background - second sentence is an incomplete sentence

Results - in the sentence discussing primary surgery - what two groups are being compared?

is completion response the same as complete response? Confusing to change the verbiage

Conclusion - mCRC is used without being previously defined in Abstract

Materials and Methods

"if the primary tumor had been operated" is not correct grammar

Abbreviations are used inconsistently throughout the article. If you are using CT to represent chemotherapy, it should be that way through the entire article

Overall, the introductions, methods, and results are much improved (and interesting) from the prior version.

The Discussion, however, remains disorganized and confusing.

The authors include information about novel anti-VEGF, anti-EGFR targeted agents, but these were not included in the study presented and it is confusing how they then tie this to their findings of longer survival times in patients who attained CR. Yes, these new agents are important to include, but I do not understand how this is related to your data?

The paragraph starting with " A total of 20,023 patients" seems completely out of order within the discussion and is not clear what first 24 clinical trials are being referenced?

Then the discussion switches back to talking about chemotherapy again.

I suggest that the authors reorganize the discussion section and ensure that they focus on their findings.

Conclusion

What results can physicians use when counseling patients with CR?

7. PLOS authors have the option to publish the peer review history of their article (what does this mean?). If published, this will include your full peer review and any attached files.

Reviewer #1: No

---

## [Author Response · Author response to Decision Letter 1]

1 May 2021

Journal Requirements:

Response th Editor: Wanted changes in references was done.

Reviewer's Responses to Questions

Comments to the Author

1. If the authors have adequately addressed your comments raised in a previous round of review and you feel that this manuscript is now acceptable for publication, you may indicate that here to bypass the “Comments to the Author” section, enter your conflict of interest statement in the “Confidential to Editor” section, and submit your "Accept" recommendation.

Reviewer #1: (No Response)

2. Is the manuscript technically sound, and do the data support the conclusions?

Reviewer #1: Yes

3. Has the statistical analysis been performed appropriately and rigorously?

Reviewer #1: Yes

4. Have the authors made all data underlying the findings in their manuscript fully available?

Reviewer #1: Yes

4. Have the authors made all data underlying the findings in their manuscript fully available?

Reviewer #1: Yes

5. Is the manuscript presented in an intelligible fashion and written in standard English?

Response the review: the article was editted for standart English 

6. Review Comments to the Author

Reviewer #1: Informed consent was obtained from all 222 patients in the study? This is not typical of a retrospective chart review.

Response the reviewer: Informed consent was obtained as it was the request of the university ethics committee. However, this information in the article has been removed to avoid confusion for the reader.

There are still multiple grammatical and typographical errors throughout the paper that make it somewhat difficult to read, for example (this does not include all the errors in the article, just a sampling):

Abstract

Background - second sentence is an incomplete sentence

Response the reviewer: second sentence was completed

Results - in the sentence discussing primary surgery - what two groups are being compared?

is completion response the same as complete response? Confusing to change the verbiage

 Response the reviewer: Result section in abstract was re-written according to reviewer recommendation.

Conclusion - mCRC is used without being previously defined in Abstract 

Response the reviewer: mCRC was defined in Abstract

Materials and Methods

"if the primary tumor had been operated" is not correct grammar

Response the reviewer: this sentence was written correctly

Abbreviations are used inconsistently throughout the article. If you are using CT to represent chemotherapy, it should be that way through the entire article

Response the reviewer: the abbreviations were written consistently by scanning the entire article.

Overall, the introductions, methods, and results are much improved (and interesting) from the prior version.

Response the reviewer: Thank you very much for your valuable contribution.

The Discussion, however, remains disorganized and confusing.

The authors include information about novel anti-VEGF, anti-EGFR targeted agents, but these were not included in the study presented and it is confusing how they then tie this to their findings of longer survival times in patients who attained CR. Yes, these new agents are important to include, but I do not understand how this is related to your data?

The paragraph starting with " A total of 20,023 patients" seems completely out of order within the discussion and is not clear what first 24 clinical trials are being referenced?

Then the discussion switches back to talking about chemotherapy again.

I suggest that the authors reorganize the discussion section and ensure that they focus on their findings. 

Changed paragraph places so that it can continue within a flow, and focus on our findings

Response the reviewer: Discussion section was re-written. extra information of except findingswas removed. Targeted agents was mentioned only in the times of overall survey in discussion.

Conclusion

What results can physicians use when counseling patients with CR?

Response the reviewer: Wanted changes was done in conclusion.

---

## [Decision Letter · Decision Letter 2]

27 May 2021

PONE-D-20-28477R2

Contribution of “Complete Response to Treatment” to Survival in Patients with Unresectable Metastatic Colorectal Cancer; A Retrospective Analysis

PLOS ONE

Dear Dr. Bulut,

Thank you for submitting your manuscript to PLOS ONE. After careful consideration, we feel that it has merit but does not fully meet PLOS ONE’s publication criteria as it currently stands. Therefore, we invite you to submit a revised version of the manuscript that addresses the points raised during the review process.

***General comments:****The authors did a big job in improving the language, the text is now better readible and understandable. Anyway there are still some uncertainties, what the sentence means.**Authors referred about metastatic colorectal cancer of two types, liver and bone metastasis. Liver metastasis is the most frequent, opposit is bone metastasis, it is one of rare metastatic site. Is some reason, why authors selected those two types? **The authors enrolled 222 patients and evaluated complete response  (Yes-No) in 212persons. The percentage for CR group is fitting with 212 persons (e.g. section Results first para, section Discussion, first para). Maybe I would recomend authors to simplify the situation and presented results only from those, who were evaluated from CR point. This is a main aim of the study. In this case Table 1 could be removed. **The authors omitted to mention the end of follow up period, the exact date (month/year). **Why the authors focused on two distinct group of patients? First, incidentally diagnosed metastatic CRC and second on those with developed metastasis after primary tumor resection? The second group might be associated with chemoresistance or treatment response.**128 patients had primary surgery, only 47 had adjuvant chemotherapy, please, add some comment.**18 patients had no LVI and 21 patients had no PNI -  does it mean that they have only liver or bone metastasis?  **The study belongs to rather small study, it should be mentioned in the limitation of the study.* ***Particular comments****Abstract:**last sentence:  ......and the combination therapy regimens...* *Material and Methods**to add the date of termination of follow up* *Results**As mentioned above, Table 1 could be removed, the text and next table contain all information.**Table 3 is also perfectly described in the text and can be omitted. Also because the numbers for Exp(B) in case of bone metastasis are nonsense and numbers in case of fluoropyrimidine are missing.**In the text is abbreviation HR, in first appearance maybe should be explained (hazard ratio). * *Table 6 - there are some empty boxes(cells) in the table, but the significance is calculated. It is confusing. Especially in the case, when is significant result (First line CT(Fluoropyrimidine)). **The abbreviation "LAP" is not explain* *Discussion**First paragraph, last sentence.**...first line CT (combination compare with fluoropyrimidine alone)...**Second paragraph: The sentence "Significantly, OS and PFS outliers were lower with increasing age [13]." is not understandable. Does it mean: "OS and PFS number of outliers was significantly lower with increasing age" ?**Fourth para, fourth sentence: my suggestion how to reformulate:**In the study of D´Angelica et al. on 49 colorectal cancer patients with unresectable hepatic metastasis was reported that only.....**Last para**Interestingly, the CR was an independent predictor of survival for about 40months in OS and for about 15 months in PFS.* *Fig. 1**Should be probably "First-line CT (combination with fluoropytimidine)" instead of "First-line CT (combination according to fluoropyrimidine)"* 

We look forward to receiving your revised manuscript.

Kind regards,

Ludmila Vodickova, M.D., PhD

Academic Editor

PLOS ONE

Journal Requirements:

Reviewers' comments:

Reviewer's Responses to Questions

**Comments to the Author**

1. If the authors have adequately addressed your comments raised in a previous round of review and you feel that this manuscript is now acceptable for publication, you may indicate that here to bypass the “Comments to the Author” section, enter your conflict of interest statement in the “Confidential to Editor” section, and submit your "Accept" recommendation.

Reviewer #2: (No Response)

2. Is the manuscript technically sound, and do the data support the conclusions?

Reviewer #2: No

3. Has the statistical analysis been performed appropriately and rigorously? 

Reviewer #2: N/A

4. Have the authors made all data underlying the findings in their manuscript fully available?

Reviewer #2: No

5. Is the manuscript presented in an intelligible fashion and written in standard English?

Reviewer #2: No

6. Review Comments to the Author

Reviewer #2: I received this manuscript after the first revision. Even though the article has been revised, my feelings are confused. The article focused on determining the complete response in the metastatic CRC. However, everything that the authors studied in their study has long been known, and the conclusions of the study do not contribute much to the already acquired knowledge about the survival of metastatic CRC patients.

The authors stated that CR was less common in the presence of liver metastasis and bone metastasis. My question is: If authors focused on mCRC and no lung or other distant metastasis were observed, what is it outcome of this observation?

It is not clear whether authors focused on sporadic CRC?

What was the end of the study, I mean when the follow up terminated?

Why the authors focused on two distinct group of patients? First, incidentally diagnosed metastatic CRC and second on those with developed metastasis after primary tumor resection? The second group might be associated with chemoresistance or treatment response.

Is it correct the metastasis interval? 5.93±12.81 (Mean ±SD)

The authors stated that there were 20 survivors and median patients follow up was in range 0.3 to almost 132 months. With no defined end of the study, this has no reason to show up.

If I understand correctly, 9% of those patients with no primary surgery reached CR while 42% with no primary surgery did not reach CR. This outcome is clear from Table 2 and further again elaborated in Table 3, 6 and 7. Why it can not be only supplemented by additional text only?

Besides, 128 patients had primary surgery however only 47 had adjuvant therapy. Why? This should be mentioned. What was the median time to develop metastasis in these patients?

Further, 18 patients had no LVI and 21 no PNI, are there really all patients in the study in stage IV?

The main limitation is also relatively small population size that might results many of outcomes as false positives. This should be commented.

Discussion is wordy and not focused on obtained results.

In addition, the article is written in very poor English and the several sentences do not make sense.

The article contains unnecessarily many tables and the following text, which comment on everything that can be found in the table and usually comments on insignificant results. Tables 3, 4 and 5 can be omitted and only stated by additional text as it is already.

7. PLOS authors have the option to publish the peer review history of their article (what does this mean?). If published, this will include your full peer review and any attached files.

Reviewer #2: No

---

## [Author Response · Author response to Decision Letter 2]

20 Jun 2021

İPONE-D-20-28477 Contribution of “Complete Response to Treatment” to Survival in Patients with Unresectable Metastatic Colorectal Cancer; Retrospective Analysis PLOS ONE  Dear Dr. Editor and Reviewers;

  

General comments:

Authors referred about metastatic colorectal cancer of two types, liver and bone metastasis. Liver metastasis is the most frequent, opposit is bone metastasis, it is one of rare metastatic site. Is some reason, why authors selected those two types? 

1-6 metastatic sites were evaluated and included in the article. however, since these two metastatic sites are very significant, the study focused on these two sites. This comment has been added to the article.

The authors enrolled 222 patients and evaluated complete response  (Yes-No) in 212persons. The percentage for CR group is fitting with 212 persons (e.g. section Results first para, section Discussion, first para). 

the number of patients is given in the article. Complete response was evaluated only in these 202 patients, as the full data of 202 patients were available.This comment has been added to the article.

Maybe I would recomend authors to simplify the situation and presented results only from those, who were evaluated from CR point. This is a main aim of the study. In this case Table 1 could be removed. 

table 1 could be removed. 

The authors omitted to mention the end of follow up period, the exact date (month/year). 

The end of follow up date (month/year) was added the manuscript

Why the authors focused on two distinct group of patients? First, incidentally diagnosed metastatic CRC and second on those with developed metastasis after primary tumor resection? The second group might be associated with chemoresistance or treatment response.

The patient group is metastatic colorectal cancer patients. This group includes patients who have previously been operated and/or received adjuvant therapy.This comment has been added to the article. 

128 patients had primary surgery, only 47 had adjuvant chemotherapy, please, add some comment.

80 patients had primary surgery but only 48 patients received adjuvant therapy. This information has also been added to the article.

18 patients had no LVI and 21 patients had no PNI -  does it mean that they have only liver or bone metastasis?  

Since the initial pathology was evaluated, there was no LVI and PNI from patients who were not metastatic at the time of diagnosis. Added as a comment to the article.

The study belongs to rather small study, it should be mentioned in the limitation of the study.

This knowledge added to the limitations paragraph.

Particular comments

Abstract:

last sentence:  ......and the combination therapy regimens…

corrected in the article.

Material and Methods

to add the date of termination of follow up

corrected in the article.

Results

As mentioned above, Table 1 could be removed, the text and next table contain all information.

Table 3 is also perfectly described in the text and can be omitted. Also because the numbers for Exp(B) in case of bone metastasis are nonsense and numbers in case of fluoropyrimidine are missing.

In the text is abbreviation HR, in first appearance maybe should be explained (hazard ratio). 

Table 1 and 3 was removed in manuscript

Table 6 - there are some empty boxes(cells) in the table, but the significance is calculated. It is confusing. Especially in the case, when is significant result (First line CT(Fluoropyrimidine)). 

The abbreviation "LAP" is not explain

corrected in the article.

Discussion

First paragraph, last sentence.

...first line CT (combination compare with fluoropyrimidine alone)...

Second paragraph: The sentence "Significantly, OS and PFS outliers were lower with increasing age [13]." is not understandable. Does it mean: "OS and PFS number of outliers was significantly lower with increasing age" ?

corrected in the article.

Fourth para, fourth sentence: my suggestion how to reformulate:

In the study of D´Angelica et al. on 49 colorectal cancer patients with unresectable hepatic metastasis was reported that only…..

corrected in the article.

Last para

Interestingly, the CR was an independent predictor of survival for about 40months in OS and for about 15 months in PFS.

corrected in the article.

Fig. 1

Should be probably "First-line CT (combination with fluoropytimidine)" instead of "First-line CT (combination according to fluoropyrimidine)”

corrected in the article.

 Journal Requirements:

Please review your reference list to ensure that it is complete and correct. If you have cited papers that have been retracted, please include the rationale for doing so in the manuscript text, or remove these references and replace them with relevant current references. Any changes to the reference list should be mentioned in the rebuttal letter that accompanies your revised manuscript. If you need to cite a retracted article, indicate the article’s retracted status in the References list and also include a citation and full reference for the retraction notice.  We review our reference list    Reviewers' comments:  Reviewer's Responses to Questions

Comments to the Author  

Reviewer #2: I received this manuscript after the first revision. Even though the article has been revised, my feelings are confused. The article focused on determining the complete response in the metastatic CRC. However, everything that the authors studied in their study has long been known, and the conclusions of the study do not contribute much to the already acquired knowledge about the survival of metastatic CRC patients. 

Sorry for your comment. There are very few studies on complete response.

I have been waiting for a year for the article to be published in your journal. I'm starting to think I've been lingering for a long time

The authors stated that CR was less common in the presence of liver metastasis and bone metastasis. My question is: If authors focused on mCRC and no lung or other distant metastasis were observed, what is it outcome of this observation? 

In this study, all metastatic areas were examined. It was evaluated from 1 to 6 metastatic sites. but there was significant results in both bone and liver. At the same time, the importance of the number of metastatic sites was mentioned in the study.

It is not clear whether authors focused on sporadic CRC? 

Patients who developed both denovo metastatic and follow-up metastases were included in the study.

What was the end of the study, I mean when the follow up terminated?  At the end of the study, patients with complete responses were found to have a long survey.

Why the authors focused on two distinct group of patients? 

These are not two different groups, they are all metastatic patients; whether it is metastasis at the time of diagnosis or metastasis develops during follow-up.

First, incidentally diagnosed metastatic CRC and second on those with developed metastasis after primary tumor resection? 

The second group might be associated with chemoresistance or treatment response. 

Added as a comment to the article.

Is it correct the metastasis interval? 5.93±12.81 (Mean ±SD) 

This is correct.

The authors stated that there were 20 survivors and median patients follow up was in range 0.3 to almost 132 months. With no defined end of the study, this has no reason to show up. If I understand correctly, 9% of those patients with no primary surgery reached CR while 42% with no primary surgery did not reach CR. 

The importance of primary surgery was mentioned in many places in the article.

This outcome is clear from Table 2 and further again elaborated in Table 3, 6 and 7. Why it can not be only supplemented by additional text only?

Table 1 and 3 was removed.

 Besides, 128 patients had primary surgery however only 47 had adjuvant therapy. Why? T his should be mentioned. 

Added as a comment to the article.

Further, 18 patients had no LVI and 21 no PNI, are there really all patients in the study in stage IV?

Added as a comment to the article.’ Because only devo metastatic patients were not evaluated’ The main limitation is also relatively small population size that might results many of outcomes as false positives. This should be commented.

Added as a comment to the article.

 Discussion is wordy and not focused on obtained results. In addition, the article is written in very poor English and the several sentences do not make sense. The article contains unnecessarily many tables and the following text, which comment on everything that can be found in the table and usually comments on insignificant results. Tables 3, 4 and 5 can be omitted and only stated by additional text as it is already.

The article’s english was edited

---

## [Editor Report · Decision Letter 3]

19 Aug 2021

PONE-D-20-28477R3

Contribution of “Complete Response to Treatment” to Survival in Patients with Unresectable Metastatic Colorectal Cancer; A Retrospective Analysis

PLOS ONE

Dear Dr. Bulut,

Thank you for submitting your manuscript to PLOS ONE. After careful consideration, we feel that it has merit but does not fully meet PLOS ONE’s publication criteria as it currently stands. Therefore, we invite you to submit a revised version of the manuscript that addresses the points raised during the review process.

The authors did great job in improving the paper, but there are still some very small things to consider and repair:

ABSTRACT:

  In addition, to is evaluate progression-free survival…– should be: In addition, to evaluate progression-free survival…

… In subgroup patients with who underwent primary surgery; the number….should be: In subgroup patients who underwent primary surgery, the number…

RESULTS: 

The sentence: Because only devo metastatic patients were not evaluated; should be deleted

Twenty (9.0%) patients were still alive, 222 (91.0%) patients had died .- it is some mistake or typing error, this does not correspond with total number of enrolled patients (222=100%). 

PFS analysis, based on whether a CR was attained is shown in Table 5 

should be . PFS analysis, based on whether a CR was attained is shown in Table 3

Table 4: 

there are missing values for Exp(B) and 95%CI in the case of: Primary region (Rectum). Histology (Adeno), Differentiation (Good) – all of them are not significant values, it is not so critical, but there are missing also in the case of First-line CT (Fluoropyrimidine), which are significant P=0.007.

Row with T0 has no sense.

First-line CT (Combination according to Fluoropyrimidine) _ I guess should be: (Combination with Fluoropyrimidine)

Table 5

First-line CT (Combination according to Fluoropyrimidine) _ I think it should be: (Combination with Fluoropyrimidine).

DISCUSSION:

The sentence: However, the vast majority (80-90%) present with unresectable diseases should be:

However, the vast majority (80-90%) presents unresectable diseases

The sentence:

…..whereas it is approximately 2 years in those treated with a combination of irinotecan or oxaliplatin to fluoropyrimidine-based CT.  

Should be:… whereas it is approximately 2 years in those treated with a combination of irinotecan or oxaliplatin with fluoropyrimidine-based CT.

FIGURE 3:

First-line CT (Combination according to Fluoropyrimidine) _ should be: Combination with Fluoropyrimidine

Decreases in Number of metastatik lesion 

should be: Decreases in Number of metastatic lesions

We look forward to receiving your revised manuscript.

Kind regards,

Ludmila Vodickova, M.D., PhD

Academic Editor

PLOS ONE

Journal Requirements:

Please clarify the nature of the informed consent in your Ethics Statement and Methods section. When and how did patients provide their consent? Did patients consent to the medical treatment, and/or did they specifically consent to participate in this study or to have their medical records used in research?
---

## [Author Response · Author response to Decision Letter 3]

25 Aug 2021

Dear Editor,  ABSTRACT:   In addition, to is evaluate progression-free survival…– should be: In addition, to evaluate progression-free survival…  … In subgroup patients with who underwent primary surgery; the number….should be: In subgroup patients who underwent primary surgery, the number…

*the necessary changes was made.

RESULTS:  The sentence: Because only devo metastatic patients were not evaluated; should be deleted

Twenty (9.0%) patients were still alive, 222 (91.0%) patients had died .- it is some mistake or typing error, this does not correspond with total number of enrolled patients (222=100%). 

Twenty (9.0%) patients were still alive, 202 (91.0%) patients had died *the necessary changes was made.

PFS analysis, based on whether a CR was attained is shown in Table 5  should be . PFS analysis, based on whether a CR was attained is shown in Table 3 *the necessary changes was made.

Table 4:  there are missing values for Exp(B) and 95%CI in the case of: Primary region (Rectum). Histology (Adeno), Differentiation (Good) – all of them are not significant values, it is not so critical, but there are missing also in the case of First-line CT (Fluoropyrimidine), which are significant P=0.007. Row with T0 has no sense. 

 Since the number of cases in the sub group is insufficient, the values mentioned in the table cannot be given. The number of patients with a complete response in the first-line CT (Fluoropyrimidine) group is only “1”.

This information has been added as a table description.

The missing values in the table was filled and Row with T0 was deleted

First-line CT (Combination according to Fluoropyrimidine) _ I guess should be: (Combination with Fluoropyrimidine) *the necessary changes was made.

Table 5 First-line CT (Combination according to Fluoropyrimidine) _ I think it should be: (Combination with Fluoropyrimidine). *the necessary changes was made.

DISCUSSION: The sentence: However, the vast majority (80-90%) present with unresectable diseases should be: However, the vast majority (80-90%) presents unresectable diseases *the necessary changes was made.

The sentence: …..whereas it is approximately 2 years in those treated with a combination of irinotecan or oxaliplatin to fluoropyrimidine-based CT.   Should be:… whereas it is approximately 2 years in those treated with a combination of irinotecan or oxaliplatin with fluoropyrimidine-based CT. *the necessary changes was made.

FIGURE 3: First-line CT (Combination according to Fluoropyrimidine) _ should be: Combination with Fluoropyrimidine Decreases in Number of metastatik lesion  should be: Decreases in Number of metastatic lesions

*the necessary changes was made.

---

## [Editor Report · Decision Letter 4]

25 Oct 2021

Contribution of “Complete Response to Treatment” to Survival in Patients with Unresectable Metastatic Colorectal Cancer; A Retrospective Analysis

PONE-D-20-28477R4

Dear Dr. Bulut,

We’re pleased to inform you that your manuscript has been judged scientifically suitable for publication and will be formally accepted for publication once it meets all outstanding technical requirements.

Kind regards,

Ludmila Vodickova, M.D., PhD

Academic Editor

PLOS ONE

---

## [Editor Report · Acceptance letter]

29 Oct 2021

PONE-D-20-28477R4 

Contribution of “Complete Response to Treatment” to Survival in Patients with Unresectable Metastatic Colorectal Cancer: A Retrospective Analysis 

Dear Dr. Bulut:

I'm pleased to inform you that your manuscript has been deemed suitable for publication in PLOS ONE. Congratulations! Your manuscript is now with our production department. 

Kind regards, 

on behalf of

Dr. Ludmila Vodickova 

Academic Editor

PLOS ONE